# Frequency, complications, and mortality of inhalation injury in burn patients: A systematic review and meta-analysis protocol

**Juliana Elvira Herdy Guerra Avila**[1]*, **Levy Aniceto Santana**[2], **Denise Rabelo Suzuki**[3], **Vinícius Zacarias Maldaner da Silva**[4], **Marcio Luís Duarte**[5], **Aline Mizusaki Imoto**[6], **Fábio Ferreira Amorim**[7]

1 Culdade de Ciências de Saúde - Universidade de Brasília-UnB, Programa de Pós-Graduação em Ciências da Saúde, FaBrasilia (DF), Brazil, 2 Programa de Pós-Graduação em Ciências da Saúde, Escola Superior de Ciências da Saúde (ESCS), Brasilia (DF), Brazil, 3 Programa de Pós-Graduação em Ciências da Saúde, Coordenação de Cursos Pós-Graduação Stricto Sensu, Escola Superior de Ciências da Saúde (ESCS), Brasilia (DF), Brazil, 4 Universidade de Brasília, Brasilia (DF), Brazil and Programa de Pós Graduação em Ciências do Movimento Humano e Reabilitação, Universidade Evangélica de Goiás, Goiás, Brazil, 5 Radiology Professor of Universidade de Ribeirão Preto, Campus Guarujá, Guarujá-SP, Brazil, 6 Programa de Pós-Graduação em Ciências da Saúde, Coordenação de Cursos Pós-Graduação Stricto Sensu, Escola Superior de Ciências da Saúde (ESCS), Brasilia (DF), Brazil, 7 Programa de Pós-Graduação em Ciências da Saúde, Coordenação de Pesquisa e Comunicação Científica, Escola Superior de Ciências da Saúde (ESCS), Brasilia (DF), Brazil

* julianabsb2010@hotmail.com

## Abstract

### Introduction

Burns are tissue traumas caused by energy transfer and occur with a variable inflammatory response. The consequences of burns represent a public health problem worldwide. Inhalation injury (II) is a severity factor when associated with burn, leading to a worse prognosis. Its treatment is complex and often involves invasive mechanical ventilation (IMV). The primary purpose of this study will be to assess the evidence regarding the frequency and mortality of II in burn patients. The secondary purposes will be to assess the evidence regarding the association between IIs and respiratory complications (pneumonia, airway obstruction, acute respiratory failure, acute respiratory distress syndrome), need for IMV and complications in other organ systems, and highlight factors associated with IIs in burn patients and prognostic factors associated with acute respiratory failure, need for IMV and mortality of II in burn patients.

### Methods

This is a systematic literature review and meta-analysis, according to the Preferred Reporting Items for Systematic Reviews and Meta-analysis (PRISMA). PubMed/MEDLINE, Embase, LILACS/VHL, Scopus, Web of Science, and CINAHL databases will be consulted without language restrictions and publication date. Studies presenting incomplete data and patients under 19 years of age will be excluded. Data will be synthesized through continuous (mean and standard deviation) and dichotomous (relative risk) variables and the total

**Data Availability Statement:** The identified research data will be made publicly available when the study is completed and published.

**Funding:** The authors received funding from Fundação de Ensino e Pesquisa em Ciências da Saúde - FEPECS, Address: SMHN 03 - conjunto A - bloco 1 - Edifício FEPECS CEP: 70701-907.

**Competing interests:** The authors have declared that no competing interests exist.

number of participants. The means, sample sizes, standard deviations from the mean, and relative risks will be entered into the Review Manager web analysis software (The Cochrane Collaboration).

## Discussion

Despite the extensive experience managing IIs in burn patients, they still represent an important cause of morbidity and mortality. Diagnosis and accurate measurement of its damage are complex, and therapies are essentially based on supportive measures. Considering the challenge, their impact, and their potential severity, IIs represent a promising area for research, needing further studies to understand and contribute to its better evolution.

The protocol of this review is registered on the International prospective register of systematic reviews platform of the Center for Revisions and Disclosure of the University of York, United Kingdom (https://www.crd.york.ac.uk/prospero), under number RD42022343944.

## Introduction

Burns are tissue traumas caused by energy transfer (thermal, chemical, electrical, radiation) [1,2] and occur with variable local and systemic inflammatory responses according to the intensity, location, and the affected area depth [3]. Due to the severity of their conditions, most patients require treatment in specialized units with intensive support and monitoring [4]. The consequences of burns represent a public health problem, ranging from physical incapacity and psychological and social damage to death [4]. As per a World Health Organization Fact Sheet dated October 2023, there are over 11,000,000 cases worldwide annually, resulting in 180,000 deaths [5]. Only in Brazil, from 2015 to 2020, there were 19,772 deaths from burns, as delineated by data provided by the Brazilian Department of Health [6]. According to US statistics from the National Inpatient Sample and the National Burn Repository, 40,000 hospitalizations are estimated yearly due to burns in the United States, with about 5% presenting inhalation injuries (IIs) [6,7]. Approximately 33% of all burn patients will require invasive mechanical ventilation (IMV), which increases significantly with II [8].

The diagnosis of respiratory system involvement is essentially clinical and can be complemented by bronchoscopy and other radiological and laboratory tests [9]. In ideal conditions, bronchoscopy should be performed in the first 24 hours in all patients with a history of smoke inhalation and is considered the gold standard for this type of evaluation [10,11]. When present, IIs significantly impact patient outcomes, increasing fluid needs during resuscitation, pulmonary complications, and mortality [12–14], serving as a marker of severity and an independent risk factor for death [13,14], especially in patients with over 20% of body surface area burned [15,16]. Besides, contrary to the recent advancements in the treatment of cutaneous burn injuries, the complex treatment of the IIs remains a challenging frontier since the pathophysiology is not fully understood, the diagnostic criteria remain unclear, the interventions are often ineffective, and the mortality remains high [17–20].

The treatment of IIs is traditionally performed through respiratory support with 100% oxygen, hyperbaric oxygen therapy, and/or protective IMV [9,21,22]. However, questions regarding the best way to identify and classify respiratory tract involvement, whether all patients should be intubated and receive IMV, which IMV mode is best indicated, and issues related to systemic toxicity are essential points that must be better elucidated [21].

In this context, the primary purpose of this study will be to assess the evidence regarding the frequency and mortality of II in burn patients. The secondary purposes will be to assess the evidence regarding the association between IIs and respiratory complications (pneumonia, airway obstruction, acute respiratory failure, acute respiratory distress syndrome—ARDS), need for IMV and complications in other organ systems, and highlight factors associated with IIs in burn patients and prognostic factors associated with acute respiratory failure, need for IMV and mortality of II in burn patients.

## Materials and methods

### Study design

This systematic literature review will be guided and reported according to the guidelines of the Preferred Reporting Items for Systematic Reviews and Meta-analysis (PRISMA) [23] (S1 Checklist). The protocol of this review is registered on the International prospective register of systematic reviews platform of the Center for Revisions and Disclosure of the University of York, United Kingdom (https://www.crd.york.ac.uk/prospero), under number RD42022343944.

### Research question

The question guiding this study will be: what is the frequency and mortality of inhalation injuries in burn patients?

The PICOS criterion was followed, where:

P (population) = burn patients;

I (exposure) = smoke inhalation;

C (comparison/control) = no smoke inhalation;

O (outcomes): frequency, mortality, need for IMV, complications;

S (study design): observational studies, clinical trials

### Inclusion criteria

**Population of interest.**   Adult patients of both sexes victims of acute burns regardless of magnitude or cause.

**Exposure type.**   Inhalation injury associated with the burn event. Inhalation injury will be defined as the damage inflicted to the respiratory tract or lung tissue from smoke, heat, and/or chemical irritants introduced into the airway during a burn event [24]. Although bronchoscopy may be performed to confirm the diagnosis of II and is considered the gold standard for this type of evaluation, the studies that applied only clinical criteria or used imaging or laboratory findings for II diagnosis will be included in the review [10,11].

**Control group.**   Adult patients victims of burns who have not been exposed to smoke inhalation.

**Outcomes evaluated.**   The primary outcomes will be:

• Frequency;

• Mortality.

Secondary outcomes will be:

- Respiratory complications: pneumonia, airway obstruction, acute respiratory failure, and ARDS;

- Need for IMV;

- Complications in other organ systems;

- Factors associated with IIs, need for IMV, complications in other organ systems, and mortality.

The ARDS Berlin definition will be used to diagnose the ARDS [25].

**Type of study included.** Observational studies and clinical trials that evaluated the frequency and mortality of burn patients exposed to smoke inhalation.

## Exclusion criteria

Studies in patients under 19 years of age will be excluded.

Studies presenting incomplete data, reviews, case series, case reports, and editorials will be excluded. Letters to the editor that do not report results from original data will also be excluded.

The inclusion and exclusion criteria are summarized in Table 1

## Methods for identification of studies

**Databases.** The search for studies will be performed without language and publication date restrictions in the following databases:

- PubMed;

- Embase;

- LILACS/VHL;

- Scopus

- Web of Science

- CINAHL

**Table 1. Inclusion and exclusion criteria.**

| Evaluation | Inclusion criteria | Exclusion criteria |
|---|---|---|
| **Population of interest** | Adult patients of both sexes victims of acute burns regardless of magnitude or cause. | Patients under 19 years |
| **Exposure type** | Inhalation injury | - |
| **Control group** | Not exposed to smoke inhalation. | - |
| **Outcomes** | Primary: Frequency and mortality. Secondary: Respiratory complications, need for IMV, complications in other organ systems, factors associated with IIs, and factors associated with acute respiratory failure, need for IMV, and mortality. | - |
| **Type of study** | Observational studies and clinical trials | Reviews, case series, case reports, and editorials. Letters to the editor that do not report results from the original study. |

IMV: Invasive mechanical ventilation.

**Search strategy.** In the search, descriptors previously identified in DeCS (Descriptors in Health Sciences, http://decs.bvs.br/), MeSH (Medical Subject Headingshttps://www.nlm.nih.gov/mesh/meshhome.html, https://www.nlm.nih.gov/mesh/meshhome.html), and Entree terms (https://www.embase.com) will be used, and their respective synonyms to include the largest number of relevant studies.

In this context, the search terms used will be:

(1) Burns, inhalation; inhalation burns; smoke inhalation injury; burn, inhalation; inhalation burn; smoke inhalation injury; inhalation injury, smoke; injury, smoke inhalation; inhalation injuries, smoke; injuries, smoke inhalation; smoke inhalation injuries; lung burn; queimaduras por inalação; quemaduras por inhalación; brûlures par inhalation; lesão por inalação de fumaça; smoke; lesión por inhalación de humo; lésion par inhalation de fumée;

(2) Epidemiology; epidemiology or incidence or prevalence or occurrence; social epidemiology; epidemiologies, social; epidemiology, social; social epidemiologies; epidemiologia; epidemiology; epidemiología; épidémiologie; epidemiologia social.

The complete search strategy for all databases is shown in Table 2.

**Table 2. Search strategy.**

| Database | Search |
|---|---|
| PUBMED/ MEDLINE | #1: "Burns, Inhalation" [MeSH] OR (Inhalation Burns) OR (Burn, Inhalation) OR (Inhalation Burn)<br>#2: "Smoke Inhalation Injury" [MeSH] OR (Inhalation Injury, Smoke) OR (Injury, Smoke Inhalation) OR (Inhalation Injuries, Smoke) OR (Injuries, Smoke Inhalation) OR (Smoke Inhalation Injuries)<br>#3: "Epidemiology" [MeSH] OR (Social Epidemiology) OR (Epidemiologies, Social) OR (Epidemiology, Social) OR (Social Epidemiologies)<br>#4: #1 OR #2 AND #3 |
| EMBASE | #1: "Lung burn" exp: explode all trees<br>#2: "Epidemiology"[MeSH]: explode all trees<br>#3: #1 AND #2 |
| LILACS / BVS | #1: mh:" Queimaduras por Inalação" OR (Burns, Inhalation) OR (Quemaduras por Inhalación) OR (Brûlures par inhalation) OR (mh:C26.200.322)<br>#2: mh:"Lesão por Inalação de Fumaça" OR (Smoke Inhalation Injury) OR (Lesión por Inhalación de Humo) OR (Lésion par inhalation de fumée) OR (mh: C26.200.322.800)<br>#3: mh:"Epidemiologia" OR (Epidemiology) OR (Epidemiología) OR (Épidémiologie) OR (Epidemiologia Social) OR (mh: H02.403.720.500) OR (mh: SP5.001) OR (mh: SP8.946.702.667.586)<br>4: #1 OR #2 AND #3 |
| SCOPUS | #1: "Burns, Inhalation" OR (Inhalation Burns) OR (Burn, Inhalation) OR (Inhalation Burn)<br>#2: "Smoke Inhalation Injury" OR (Inhalation Injury, Smoke) OR (Injury, Smoke Inhalation) OR (Inhalation Injuries, Smoke) OR (Injuries, Smoke Inhalation) OR (Smoke Inhalation Injuries)<br>#3: "Lung burn"<br>#4: "Epidemiology" OR (Social Epidemiology) OR (Epidemiologies, Social) OR (Epidemiology, Social) OR (Social Epidemiologies)<br>#5: #1 OR #2 OR #3 AND #4 |
| Web of Science | #1: "Burns, Inhalation": explode all trees<br>#2: "Smoke Inhalation Injury": explode all trees<br>#3: "Epidemiology": explode all trees<br>#4: #1 OR #2 AND #3 |
| CINAHL | #1: "Burns, Inhalation": explode all trees<br>#2: "Smoke Inhalation Injury": explode all trees<br>#3: "Epidemiology or incidence or prevalence or occurrence": explode all trees<br>#4: #1 OR #2 AND #3 |

In addition, grey literature reports will be sourced through simplified searches on Google Scholar and worldwidescience.org.

Finally, forward and backward reference searches will be performed to identify any other potential studies that might have been missed in the search process (backward and forward snowballing).

## Selection and data analysis

**Selection of studies and evaluation of methodological quality.** All references found by the searches will be organized using the Rayyan platform for Systematic Review (https:// rayyan.qcri.org/) used as a tool for removing duplicates, selecting, and screening studies. The data extraction from the selected studies, such as information from the participants and analyzed outcomes, will be performed manually using Microsoft Word.

Two reviewers will independently perform the studies' selection (JA and DS in the authors' list). The Rayyan platform provides an interface for each reviewer. Then, it indicates which studies showed disagreements in the analysis so that a third reviewer (AI in the authors' list) can resolve them.

Initially, the title and abstract will be analyzed. Next, the third reviewer (AI in the authors' list) will analyze the inclusion and exclusion disagreements about a particular study. Then, the texts will be fully evaluated, and the studies composing the review will be defined.

Again, for studies with a disagreement between the two main reviewers, the third reviewer (AI in the authors' list) will resolve the conflicts. Studies not meeting the inclusion criteria will be excluded, and the reasons for this decision will be recorded.

The nonrandomized eligible studies will be included in the risk of bias assessment stage through the Newcastle Ottawa Scale Tool [26] and randomized controlled trials through the Cochrane Risk of Bias 2 (RoB 2) tool 2019 version [27].

The study selection process will be performed based on PRISMA Flow Diagram [23].

**Data extraction process.** Data extraction will be performed according to criteria related to the following protocols:

- general characteristics of the studies (author, year, title, journal, country and language of publication, study design);

- information on participants (age, sex, ethnicity, diagnosis or specific characteristics, sample size);

- exposure data (description of inhaled material, duration of exposure);

- data on control;

- characteristics of inhalation injuries;

- data related to outcomes (frequency, mortality, need for IMV, development of complications in other organ systems).

**Summary of results and statistical analysis.** Outcome scores after the intervention will be extracted from the included studies and collected using continuous (mean and standard deviation) and dichotomous (relative risk) data and the total number of participants. When numerical data are missing, the authors will be contacted via e-mail, requesting additional data for analysis.

Means, sample sizes, standard deviations from the mean, and relative risks will be entered into the Review Manager analysis software, version 5.3 (The Cochrane Collaboration), which

will be used to quantify the results. Statistical significance will be defined as $p < 0.05$. Since the outcomes of interest will be evaluated with different scales and units, standardized measurements will be used to calculate the effect sizes, standard mean deviation, and 95% confidence intervals (95% CI).

For further comparisons concerning the extent of burn injury size, shock, or presence of infections, subgroup analysis will be performed if feasible.

## Assessment of risk of bias

For this evaluation, the Newcastle Ottawa Scale tool will be used for the nonrandomized studies [26] and the RoB 2 (2019) version for the randomized controlled trials [27]. Two reviewers will independently evaluate the risk of bias of the included studies (JA and DS in the authors' list). The third reviewer (AI in the authors' list) will resolve the disagreements.

## Quality of the evidence

For this evaluation, the criteria of the Grading of Recommendations Assessment, Development and Evaluation (GRADE) working group will be used [28]. GRADE assesses the quality of the evidence based on the assessment of five domains: risk of bias, imprecision, inconsistency, indirectness, and publication bias [28]. Two reviewers will independently evaluate the quality of evidence of the included studies (JA and DS in the authors' list). The third reviewer (AI in the authors' list) will resolve the disagreements.

## Ethical considerations

The research will be performed with information from studies published in electronic databases, respecting ethical principles at all stages. When processing the data collected, the principles of fidelity to the authors and respect for textual integrity will be protected.

The reviewers will not have any connection with the authors of the articles; therefore, there will be no conflicts of interest.

## Discussion

Inhalation injury is a frequent condition following burn injury, notably increasing the frequency with the rise of the burn injury size and patient age [29,30]. Although there is already extensive experience managing IIs in burn patients, they still represent a great challenge, mainly due to their complex pathophysiological process that has not yet been fully clarified, where the involvement of several inflammatory cells, mediators, and cytokines has been demonstrated [19]. Diagnosis and accurate measurement of its damage are also complex, and therapies are essentially based on supportive measures [17].

In the IIs, the magnitude and location of the injury vary considerably according to the environment and the host factors [31]. In this respect, several factors should be considered, such as ignition source, concentration and solubility of inhaled substances, diameter and size of the particles in the smoke, exposure duration, temperature, and patient immune response [20,31,32]. Individuals aged 65 and beyond exhibit a mortality rate from burns exceeding the average six factors [33]. Due to diminished physiological reserves and comorbidities, managing this demographic poses a distinctive and formidable challenge. Multiple preexisting risk factors are manifest in older adults, encompassing an elevated susceptibility to infections, pulmonary diseases, and comorbidities [34].

Although most patients exposed to smoke inhalation evolve well, the development of respiratory injury significantly worsens the outcome of these patients with a significant increase in

mortality and complications, including long-term sequelae [17,34–36]. Indeed, pulmonary complications following burns and II cause or directly contribute to 77% deaths [37,38]. Among the pulmonary complications, ARDS may develop early or several days after the exposure [39]. Although ARDS may also occur in burns without II, the clinical symptoms tend to worsen following IIs. In II, ARDS usually starts earlier, progresses with greater severity, and requires IMV for longer [13]. Furthermore, sepsis and acute respiratory failure are frequent causes of morbidity and mortality in patients with exclusive thermal burns, which may be even more prevalent in patients with IIs [13].

It is already known that II is an independent risk factor for mortality in patients with small and moderate burns [13]. In this respect, the management of IIs is essential and can vary from the conservative approach to more elaborate options involving drugs [23]. Specific treatments have been tested to prevent IMV, complications, and poor outcomes. Some studies observed that N acetylcysteine and inhaled anticoagulants (such as heparin) may effectively treat inhalation injury, significantly improving lung compliance and airway obstruction, reducing reintubation rates, increasing the number of ventilator-free days, and decreasing hospital length of stay, and mortality [40–43].

Respiratory impairment is still a major challenge in clinical practice and a promising area for research, needing further studies to understand and act on this potentially severe condition. In this systematic review, we aim to clarify the principal voids in the existing literature regarding fire-related II to guide future studies. Furthermore, the findings can contribute to diagnostic and management protocols for II in burned patients, which may improve health care and the prognosis of these people. In this aspect, especially the identification of factors associated with acute respiratory failure, need for IMV, and mortality may contribute to defining the phenotype of inhalation burns that is associated with poor prognosis and clinical approaches that may have led to better outcomes, which can contribute to stricter monitoring of these patients and the institution of earlier clinical therapeutic approaches to improve outcomes for these patients.

## Supporting information

**S1 Checklist. PRISMA-P (Preferred Reporting Items for Systematic review and Meta-Analysis Protocols) 2015 checklist: Recommended items to address in a systematic review protocol*.**
(DOC)

## Author Contributions

**Conceptualization:** Juliana Elvira Herdy Guerra Avila, Marcio Luís Duarte, Fábio Ferreira Amorim.

**Data curation:** Marcio Luís Duarte, Aline Mizusaki Imoto.

**Investigation:** Denise Rabelo Suzuki.

**Methodology:** Fábio Ferreira Amorim.

**Project administration:** Juliana Elvira Herdy Guerra Avila, Fábio Ferreira Amorim.

**Writing – original draft:** Juliana Elvira Herdy Guerra Avila.

**Writing – review & editing:** Juliana Elvira Herdy Guerra Avila, Levy Aniceto Santana, Denise Rabelo Suzuki, Vinícius Zacarias Maldaner da Silva, Marcio Luís Duarte, Aline Mizusaki Imoto, Fábio Ferreira Amorim.

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
