## [Decision Letter · Decision Letter 0]

12 Sep 2023

PONE-D-23-20903Incidence and mortality in burn patients due to inhalation injury: a systematic review and meta-analysis protocolPLOS ONE

Dear Dr. Herdy Guerra Ávila,

Thank you for submitting your manuscript to PLOS ONE. After careful consideration, we feel that it has merit but does not fully meet PLOS ONE’s publication criteria as it currently stands. Therefore, we invite you to submit a revised version of the manuscript that addresses the points raised during the review process.

We look forward to receiving your revised manuscript.

Kind regards,

Mohamed Boussarsar, M.D.

Academic Editor

PLOS ONE

Journal Requirements:

Additional Editor Comments:

The manuscript presents a protocol for a systematic review and meta-analysis on the epidemiology and burden of inhalation burns, with a focus on their association with invasive mechanical ventilation and poor prognosis.

The introduction section provides an overview of the research problem and its significance. It effectively introduces the topic of inhalation burns, emphasizing their morbidity and mortality impact worldwide. The authors contextualize the importance of this study by mentioning the global prevalence of burn cases and the substantial healthcare burden it presents. They also draw attention to the prevalence of inhalation injuries in burn patients, which is a good strategy to underscore the significance of this study.

However, this paragraph is very long and the introduction could be enhanced by discussing specific gaps in the existing literature related to the frequency of inhalation burns and their association with invasive mechanical ventilation and complications. While the authors mention that there is room for advancement in the treatment of respiratory impairments following inhalation burns, a more explicit statement about the current knowledge limitations and the clinical relevance of addressing these gaps would strengthen the introduction.

The methods section is comprehensive and well-structured. It outlines the study design, research question, inclusion and exclusion criteria, and the data sources clearly. The use of the PRISMA guidelines for conducting the systematic review and meta-analysis is appropriate and enhances the rigor of the study.

One notable strength is the registration of the protocol on PROSPERO, which adds transparency and ensures that the review is conducted following a predefined plan. The PICOS criteria are also well-defined, and the inclusion and exclusion criteria are clearly outlined. It's a robust framework for selecting studies.

However, the methods section could benefit from additional details on the search strategy, such as the specific search terms used in the databases and the strategy for manual reference searching. These details would provide transparency and aid in replication. Additionally, the methods do not explicitly mention a plan for assessing the quality of the included studies, which is a crucial aspect of systematic reviews and meta-analyses.

The authors are invited to check for the definition of "incidence" and "prevalence". I would suggest here the term "frequency" to avoid any inconsistency.

The authors are gently suggested to identify a particular phenotype of inhalation burns that is associated with invasive mechanical ventilation and poor prognosis. They may plan to define and search for exploratory factors and explanatory factors associated to this phenotype.

They may also plan for including recent studies using specific treatments such as inhaled anticoagulants to prevent invasive mechanical ventilation, complications and poor outcome.

I suggest to include some projected results with the description of different characteristics projected to be studied.

The discussion section appropriately emphasizes the significance of inhalation injuries in burn patients and the challenges associated with diagnosis and treatment. It also highlights the need for further research in this area. However, it would be beneficial to discuss the potential implications of the projected study results in more detail. For example, how might the findings impact clinical practice, patient outcomes, or future research directions.

Reviewers' comments:

Reviewer's Responses to Questions

**Comments to the Author**

1. Does the manuscript provide a valid rationale for the proposed study, with clearly identified and justified research questions?

Reviewer #1: Yes

Reviewer #2: Yes

Reviewer #3: Yes

Reviewer #4: Partly

Reviewer #5: Partly

2. Is the protocol technically sound and planned in a manner that will lead to a meaningful outcome and allow testing the stated hypotheses?

Reviewer #1: Yes

Reviewer #2: Yes

Reviewer #3: Yes

Reviewer #4: Partly

Reviewer #5: Yes

3. Is the methodology feasible and described in sufficient detail to allow the work to be replicable?

Reviewer #1: Yes

Reviewer #2: Yes

Reviewer #3: Yes

Reviewer #4: Yes

Reviewer #5: Yes

4. Have the authors described where all data underlying the findings will be made available when the study is complete?

Reviewer #1: No

Reviewer #2: Yes

Reviewer #3: Yes

Reviewer #4: Yes

Reviewer #5: Yes

5. Is the manuscript presented in an intelligible fashion and written in standard English?

Reviewer #1: Yes

Reviewer #2: Yes

Reviewer #3: No

Reviewer #4: No

Reviewer #5: Yes

6. Review Comments to the Author

You may also provide optional suggestions and comments to authors that they might find helpful in planning their study.

Reviewer #1: In this manuscript entitled "Incidence and mortality in burn patients due to inhalation injury: a systematic review and meta-analysis protocol" Id: PONE-D-23-20903, the authors present the protocol of a systematic review and meta-analysis focused on a hyper-specialized subject where this type of writing is lacking: Inhalation injury in critical burns patients. It is an interesting idea and a solid design and a clear and detailed methodology that is robust and a priori conforms to the norms for carrying out this type of study.

Nevertheless, I raised some concerns and questions:

Note of form: I consider the introduction is very long (too much epidemiology))), many parts can be switched to the discussion (such as pathophysiology of the lung injury inhalation, diagnostic tools, therapeutic modalities, etc. .)

Regarding the primary objective: the authors announce throughout the manuscript to evaluate mortality and epidemiology. For the last, sometimes it is announced incidence and sometimes prevalence. Will it depend on the type of results collected from the different studies included? Or is it just a simple proportion of lung injury by inhalation among the total number of burns counted?

Page 8, L 194: I (exposure) = inhalation of toxic substances. What kind of toxic substances is covered here: Chemical irritants, toxic metals, asphyxiants, products of fires and combustion, other? All of them? Since lung damage differs depending on the inhaled substance...

As exclusion criterion, the authors selected Studies that evaluated specific interventions in patients victims of burn exposed to inhalation of gases and toxic substances. It would be preferable if they provide more detail about this type of specific intervention and the reason why they exclude studies that are interested in it.

Relating to the 4 secondary objectives which evaluate the factors of association between: 1-inhalation injuries / need for invasive ventilation, 2- inhalation injuries / respiratory complications, 3-inhalation injuries / clinical complications and 4-factors associated with inhalation injuries:

o For the first: the attributability of the need for mechanical ventilation to the inhalation injury is not guaranteed (indeed the ARDS induced by skin burns is in itself a frequent cause of need to ventilation). Thus, the causal link the effect is far from being established in the various studies to be collected. Do the authors plan to adjust the selection criteria to minimize this bias?

o For the 4 association studies, are there predefined factors (according to the literature) that will be introduced in the analysis of each type of association?

Thank you for giving me the opportunity to review this paper

Cordially

Reviewer #2: Smoke inhalation in burn victims is associated with morbidity and mortality. A review of the literature on the incidence and complications secondary to pulmonary burns would be of great help in setting up specific therapies in pulmonary damage.

The protocol of this study is well-structured and meets the recommendations of a research study.

Comments

L 166 : objectives

178 Evaluate the factors associated with inhalation injuries in burn patients.

Replace this suggestion by " highlight prognostic factors associated with inhalation injuries in burn patients"

Reviewer #3: I read with great interest the paper titled “Incidence and mortality in burn patients due to inhalation injury: a systematic review and meta-analysis protocol”.

The rational of this protocol is captivating. However, some changes are requested in order to improve the scientific quality (ie; writing style and methods) of this protocol.

Here are my remarks classified by themes

GENERAL REMARKS

*In order the shorten the Abstract and the paper, I recommend to abbreviate

inhalation injury as II and inhalation injuries as IIs at the first use

Invasive mechanical ventilation as IMV at the first use

*Change, when appropriate, over all the paper “mechanical ventilation” by “invasive mechanical ventilation”

*The paper need to be checked by a person fluent in English (or it is possible to use ChatGPT 3.5 to correct and improve the medical writing).

*Discordance between text and table 1: is it 19 years or 1 years?

*Change gender by sex. Please see the AMA recommendation related to the use of sex and gender: https://www.ama-assn.org/press-center/press-releases/ama-adopts-new-policies-2018-interim-meeting

*Delete the abbreviations’ list

TITLE

It does not really describe the aims of the study, since only the primary objective (Incidence and mortality) was reported. I recommend (if possible) to improve the title by including the other aims.

ABSTRACT

L71: Morbidity and mortality of what? More precision is needed.

L74-76: change “It is important to recognize early and know the factors associated with this type of commitment to better conduct and thereby improve its evolution” by “It is important to recognize II early and know the factors associated with it to better conduct and thereby improve its evolution”.

L88: change as ‘variables and the total number of participants”.

KEYWORDS

Please:

Opt for MeSH terms

Do not cite as keywords some terms previously used in the title or the abstract

Classify the key words in alphabetical order

INTRODYCTION

Major remark: this important section MUST be organised into three paragraphs (not 9 as done in this paper). The first paragraph should describe what is known. The second paragraph must detail what is unknown. The last paragraph should be reserved to your aims. Therefore, I kindly ask you to reorganise your introduction.

Major remark: Several sentences are lacking references (see below)

L111: add a reference after monitoring

L112: Morbidity and mortality of what?

L113: add a reference after damage

L114: can you add the exact date (month, year) where data (ie; 11,000,000 new cases and 180,000 deaths worldwide every year, and in Brazil, there are around 1,000,000 incidents annually) were collected? Moreover add at least 2 references to argue your data

L116: the expression “of these” is related to what worldwide data or Brazilian data?

L130: Change “II represents severity..”

L132-134: correct the English style, since the sentence is unclear “In addition…….condition [14]

L157: add a reference after life support

L160: add a reference after complications

L166-178: the subsection OBJECTIVES must reintegrate the Introduction section (must be the third paragraph). Here is a proposition.

The primary objective of this study will be to evaluate the incidence and mortality in burn patients due to IIs. The secondary objectives will be to evaluate the i) Association between IIs and the need for IMV in burn patients; ii) Association between IIs and respiratory complications in burn patients; iii) Association between IIs and clinical complications related to the various organ systems in burn patients; and iv) Factors associated with IIs in burn patients.

METHODS

MAJOR POINT: avoid redundancy between text and tables 1 and 2.

L185: delete (PROSPERO)

L225: delete over 20 years of age

L240: delete (RCTs)

L254: MeSH not Mesh

L255-256: change as “Entree terms (https://www.embase.com)”

L282: over all the paper use the following model when describing the reviewers: (JA and DS in the authors’ list).

L295: add a reference after “Newcastle Ottawa Scale Tool”.

L318: how you will contact the studies' authors ?

L325: delete (SMD)

DISCUSSION

L356: add a reference after severe condition

REFERENCES

Ref 1 to 4: please cite the English titles..

TABLE 1.

Population: 19 years not 17 years, sex not gender,

Outcomes: Secondary: need for, “mechanical ventilation” by “invasive mechanical ventilation”

Type of study: should you report that only RCT will be included (better that Observational studies). In fact, there is a discordance between text and table 1 regarding this point (see L240)

Type of study: do will exclude letters to editor and editorials

Reviewer #4: Dear authors

I have read with a great interest the manuscript titled “Incidence and mortality in burn patients due to inhalation injury: a systematic review and meta-analysis protocol”.

The topic is promising and addresses a significant gap in burn injury researches.

Overall, I find the study design to be well-structured and relevant to the field of burn injury research.

However, there are a few key points that may require revision and improvement:

Majors comments

1. Scope and Focus of the study

The authors have outlined their objectives and outcomes effectively, focusing on incidence and mortality related to inhalation injury in burn patients. However, I suggest that the scope of the study could be enhanced by including the clinical presentation and the severity of patients included in the analysis. Characterizing a distinct phenotype of patients at risk could provide valuable insights into the management and prognosis of inhalation injuries. By delving deeper into patient characteristics and clinical profiles, the study can potentially offer a more comprehensive understanding of this critical medical condition. Furthermore, given the absence of a consensus on the management of these patients, it is essential to include an evaluation of different management strategies and their impact on outcomes. This addition would enhance the clinical relevance and applicability of the study's findings.

2.Inhalation injury and poor prognosis

The incrimination of inhalation injury in the prognosis of patients as complex as burns must be handled carefully by the authors, given that several confounding factors can lead to poor prognosis in those patients, such as shock, delirium, infections. In the protocol, it is crucial for the authors to outline their strategy for addressing these confounding factors. Exploring subgroups based on the severity of confounding factors, such as the extent of shock or presence of infections, can provide valuable insights into how these variables interact with inhalation injury in determining patient prognosis.

3. Definitions

In my opinion it is essential that the protocol provides comprehensive and explicit definitions for all the studied outcomes and the exploratory factors employed. Precise definitions are vital to ensure consistency in the assessment of outcomes across included studies. For instance, if "respiratory complications" is a key outcome, the protocol should specify which complications are considered, such as pneumonia, acute respiratory distress syndrome (ARDS), bronchitis, or other relevant conditions.

Likewise, the protocol should explicitly outline the criteria and methods utilized for diagnosing inhalation injury. Specify whether clinical assessment, bronchoscopy, imaging, or other diagnostic tools will be considered as valid methods for diagnosing inhalation injury. Additionally, discuss any threshold values or criteria used to define the presence or severity of inhalation injury based on these diagnostic methods.

4. Excluded studies

The exclusion of studies that evaluated specific interventions in patient victims of burns exposed to inhalation injury, should be explained by the authors.

5. English Language Quality:

The manuscript would benefit from improvement in the quality of English language usage. There are several instances of grammatical errors and awkward phrasing, throughout the document. Therefore, I recommend a thorough language edit to ensure that the manuscript is easy to read and understand.

Generals comments

1. The title formulation of the manuscript could benefit from enhancement. For instance, consider revising it to: "Incidence and Mortality of Inhalation Injuries in Burn Patients: A Systematic Review and Meta-Analysis protocol."

2. There is a confusion in the manuscript between prevalence and incidence, although I think it is actually a study of frequency.

3. There is a confusion in Age of excluded patients

Reviewer #5: I would like to extend my sincere appreciation for the opportunity to review the manuscript titled "Incidence and mortality in burn patients due to inhalation injury: a systematic review and meta-analysis protocol." It is both an honor and a privilege to contribute to the evaluation of this important work, which addresses a critical topic in the field of critical care, burn care and inhalation injury.

The significance of understanding the incidence and mortality associated with inhalation injuries in burn patients cannot be overstated, as it holds substantial implications for clinical practice and patient outcomes.

I commend the authors for their initiative in undertaking this research, and I look forward to conducting a thorough evaluation of the manuscript.

Thank you once again for entrusting me with the peer review of this manuscript, and I eagerly anticipate the opportunity to provide constructive feedback and contribute to the advancement of this vital field of study.

I have a few general comments regarding the manuscript as a whole:

I noticed the use of both “prevalence” and “incidence” separately in different sections. As this might be a source of confusion, I suggest to stick to using “the prevalence and incidence of”.

This manuscript has quite a number of grammatical mistakes and I sometimes had trouble following sentences. I think thorough copyediting is necessary.

I have a few additional reservations I found within the manuscript

In the objectives:

I find that the primary objective is clear and directly addresses a fundamental aspect. It's essential to assess the incidence and mortality associated with inhalation injuries in burn patients to understand the severity and impact of these injuries.

However, I suggest grouping and incorporating the secondary outcomes into a common comprehensive one: to identify the clinical trajectory of the most severe patients integrating the circumstances and severity of the burn, time to onset of the acute respiratory failure, the need for IMV, and the occurrence of extra respiratory end-organ failure.

To this same end, I wondered the reason for excluding patients having underwent specific interventions? I think it would have been more interesting to summarize in a direct manner the different interventions and their association to the trajectory and phenotype of the most severe patients.

You should clearly define what is considered as "respiratory complications". And more than that, specifically differentiate in studies between primary injuries associated to smoke inhalation and secondary complications such as ARDS.

In the methods section:

I noticed a discrepancy between the exclusion criteria in table I and those in the methods section (excluding those under 17 years of age in the table versus excluding those under 19). I would like to ask the authors to clarify this as I might have misunderstood the explanation behind it.

I find that this protocol is able to provide important information in the management of such patients, if a few modifications are made in the methodology with the comprehensive originality of providing the clinical trajectory and phenotype of the most severe patients, those with the highest mortality rates. This would allow the final study to identify treatable traits in this modern era of precision therapy and precision medicine.

7. PLOS authors have the option to publish the peer review history of their article (what does this mean?). If published, this will include your full peer review and any attached files.

Reviewer #1: **Yes: **Ahlem Trifi

Reviewer #2: **Yes: **Amel MOKLINE

Reviewer #3: **Yes: **Helmi BEN SAAD

Reviewer #4: **Yes: **Emna Ennouri

Reviewer #5: No

---

## [Author Response · Author response to Decision Letter 0]

31 Oct 2023

Dear Editor,

Thank you for considering the paper "Frequency, complications and mortality of inhalation injury in burn patients: a systematic review and meta-analysis protocol" by Ávila et al., for publication in the PLOS One and allowing us to re-submit this revised manuscript.

The authors are grateful to the reviewers for the careful appraisal, positive comments, and helpful criticisms. Suggested changes have been addressed. We believe that the quality of the manuscript has been improved and hope that it now meets the quality required for publication in the International PLOS One.

A detailed point-by-point response is given in a separate document.

We are looking forward to your decision regarding the suitability of the revised version of this paper for the Journal.

Thank you very much in advance,

Sincerely, 

Juliana Elvira Herdy Guerra Avila, MD, MsC*

* Corresponding author

E-mail: julianabsb2010@hotmail.com

https://orcid.org/0000-0003-2304-2046

Programa de Pós-Graduação em Ciências da Saúde, Universidade de Brasilia, Brasilia (DF), Brazil

---

## [Decision Letter · Decision Letter 1]

20 Nov 2023

Frequency, complications, and mortality of inhalation injury in burn patients: a systematic review and meta-analysis protocol

PONE-D-23-20903R1

Dear Dr. Herdy Guerra Ávila,

We’re pleased to inform you that your manuscript has been judged scientifically suitable for publication and will be formally accepted for publication once it meets all outstanding technical requirements.

Kind regards,

Mohamed Boussarsar, M.D.

Academic Editor

PLOS ONE

Additional Editor Comments (optional):

I thank the authors for their efforts in responding to my comments. The revised manuscript is now acceptable for publication. However, there are still a few minor remarks by Reviewers 3 and 5 that need to be addressed.

Reviewers' comments:

Reviewer's Responses to Questions

**Comments to the Author**

1. Does the manuscript provide a valid rationale for the proposed study, with clearly identified and justified research questions?

Reviewer #1: Yes

Reviewer #2: Yes

Reviewer #3: Yes

Reviewer #4: Yes

Reviewer #5: Yes

2. Is the protocol technically sound and planned in a manner that will lead to a meaningful outcome and allow testing the stated hypotheses?

Reviewer #1: Yes

Reviewer #2: Yes

Reviewer #3: Yes

Reviewer #4: Yes

Reviewer #5: Partly

3. Is the methodology feasible and described in sufficient detail to allow the work to be replicable?

Reviewer #1: Yes

Reviewer #2: Yes

Reviewer #3: Yes

Reviewer #4: Yes

Reviewer #5: Yes

4. Have the authors described where all data underlying the findings will be made available when the study is complete?

Reviewer #1: Yes

Reviewer #2: Yes

Reviewer #3: Yes

Reviewer #4: Yes

Reviewer #5: No

5. Is the manuscript presented in an intelligible fashion and written in standard English?

Reviewer #1: Yes

Reviewer #2: Yes

Reviewer #3: Yes

Reviewer #4: Yes

Reviewer #5: Yes

6. Review Comments to the Author

You may also provide optional suggestions and comments to authors that they might find helpful in planning their study.

Reviewer #1: the authors have adequately responded to the comments and questions I raised.

Thanks for your review. No further comments

Reviewer #2: I have read with a great interest the revised manuscript titled "Incidence and mortality in burn patients due to inhalation injury: a systematic review and meta-analysis protocol".I would like to thank the author and his collaborators for their efforts in making the necessary changes requested by the reviewers.

inhalation injury remains a great challenge for clinicians and the topic is promising and addresses a significant gap in burn injury researches. I thought that this article is ready for publication

Reviewer #3: I have carefully reviewed the revised version of the study protocol titled “Frequency, Complications, and Mortality of Inhalation Injury in Burn Patients: A Systematic Review and Meta-Analysis Protocol”.

The authors have diligently addressed all of my comments, and I commend them for their excellent work and the professional manner in which they handled my suggestions and remarks.

I highly recommend accepting this study protocol. However, I suggest that the authors make a few minor corrections (during the PDF proof for example):

- Page 6, line 125: Change " United States" to “US" as done before

- Page 8, line 182: Replace "Inhalation injury" with "II" two times.

- Page 17, line 340: Replace "inhalation injury" with "II."

- Pages 18, lines 376-377: Replace "Inhalation injury" with "II."

These adjustments will enhance the clarity and precision of the document.

Reviewer #4: I appreciate the efforts made by the authors in addressing previous concerns. I have no more additional comment.

Reviewer #5: I appreciate the efforts put into improving the manuscript. Most of my comments have been satisfyingly addressed. The English, however, still needs improvement before the manuscript becomes suitable for publication.

I have two remaining reservations that I feel were less taken into consideration.

In your response about the objectives of the study, you answered that it “will be possible to evaluate specific interventions such as factors to prevent IMV, complications, and poor outcomes”. I believe that readers would be very interested in those specific results. However, I noticed that you did not mention it in the manuscript. In fact, evaluating interventions, or at least listing interventions performed in different settings, was not mentioned throughout the text. I imagine that it would grant the study a more attractive methodology.

Moreover, I suggest defining an additional objective, with a comprehensive approach aiming to identify a phenotype of burn patients developing inhalation injuries, those most associated with mechanical ventilation and mortality. Identifying treatable trait, with specific biomarkers and treatment plan, would very much increase the value of your work.

7. PLOS authors have the option to publish the peer review history of their article (what does this mean?). If published, this will include your full peer review and any attached files.

Reviewer #1: **Yes: **Ahlem Trifi

Reviewer #2: **Yes: **Amel MOKLINE, Ph. D

Reviewer #3: **Yes: **Helmi BEN SAAD (MD, PhD)

Reviewer #4: **Yes: **Emna Ennouri

Reviewer #5: No

---

## [Editor Report · Acceptance letter]

28 Mar 2024

PONE-D-23-20903R1 

PLOS ONE

Dear Dr. Herdy Guerra Ávila, 

I'm pleased to inform you that your manuscript has been deemed suitable for publication in PLOS ONE. Congratulations! Your manuscript is now being handed over to our production team.

Kind regards, 

on behalf of

Professor Mohamed Boussarsar 

Academic Editor

PLOS ONE